# Epstein-Barr Virus Exploits the Secretory Pathway to Release Virions

**DOI:** 10.3390/microorganisms8050729

**Published:** 2020-05-13

**Authors:** Asuka Nanbo

**Affiliations:** The National Research Center for the Control and Prevention of Infectious Diseases, Nagasaki University, Nagasaki 852-8523, Japan; nanboa@nagasaki-u.ac.jp; Tel.: +81-95-819-7970

**Keywords:** Epstein-Barr virus, egress, lytic cycle, secretory pathway, Rab GTPases

## Abstract

Herpesvirus egress mechanisms are strongly associated with intracellular compartment remodeling processes. Previously, we and other groups have described that intracellular compartments derived from the Golgi apparatus are the maturation sites of Epstein-Barr virus (EBV) virions. However, the mechanism by which these virions are released from the host cell to the extracellular milieu is poorly understood. Here, I adapted two independent induction systems of the EBV lytic cycle in vitro, in the context of Rab GTPase silencing, to characterize the EBV release pathway. Immunofluorescence staining revealed that p350/220, the major EBV glycoprotein, partially co-localized with three Rab GTPases: Rab8a, Rab10, and Rab11a. Furthermore, the knockdown of these Rab GTPases promoted the intracellular accumulation of viral structural proteins by inhibiting its distribution to the plasma membrane. Finally, the knockdown of the Rab8a, Rab10, and Rab11a proteins suppressed the release of EBV infectious virions. Taken together, these findings support the hypothesis that mature EBV virions are released from infected cells to the extracellular milieu via the secretory pathway, as well as providing new insights into the EBV life cycle.

## 1. Introduction

Epstein–Barr virus (EBV) is a ubiquitous human gammaherpesvirus that infects the majority of the world population. EBV infections are persistent, but mostly asymptomatic. However, they are also associated with various lymphoid and epithelial malignancies, such as Burkitt’s lymphoma (BL), Hodgkin’s disease, gastric carcinoma, and nasopharyngeal carcinoma [1]. Similarly to other herpesviruses (alphaherpesviruses and betaherpesviruses), gammaherpesviruses depend on a recently discovered virion maturation mechanism, composed of budding and fusion events that occur in the intracellular compartments during replication [2,3,4,5,6].

During a productive infection, the newly synthesized viral genomes are packaged into capsids in the nucleoplasm, which then bud through the inner nuclear membrane into the perinuclear space, acquiring their primary envelope. This envelope is then lost by fusion with the outer nuclear membrane, leading to the release of the nucleocapsids into the cytoplasm. In the cytoplasm, these de-enveloped nucleocapsids acquire a series of tegument proteins and then their secondary envelope by budding into different cellular compartments, producing the mature virions [3]. While alphaherpesviruses bud into vesicles derived from the trans-Golgi network (TGN) [7,8,9,10,11,12,13,14] and early endosomes (EE) [15], betaherpesviruses generate unique compartments from the TGN, EE, multi-vesicular bodies, and late endosomes (LE) [6,16,17,18,19,20,21]. Although the same virion maturation mechanism was reported for gammaherpesviruses, including EBV [22,23,24,25,26,27], Kaposi sarcoma-associated herpesvirus (KSHV) [28], and Murine gammaherpesvirus 68 (MHV-68) [26], their secondary envelopment processes were mostly uncharacterized. Recently, our group showed that EBV matures by budding into compartments containing markers from both the cis-Golgi network and the TGN [29].

Ultimately, herpesviruses are released by fusion of the enveloped virions with the plasma membranes (PM). However, effective viral release depends not only on these final fusion events, but also on the transport mechanisms that allow the mature virions to reach the PM. A few studies have characterized the alphaherpesviruses release mechanism. Using live-cell fluorescence microscopy, Hogue and colleagues demonstrated that vesicles containing pseudorabies virus (PRV) were associated with small Rab GTPases, such as Rab6a, Rab8b, and Rab11a, involved in the secretory pathway-mediated vesicular transport. These vesicles frequently appeared near patches of LL5b, component of a complex that anchors microtubule plus ends to the PM, suggesting that their intracellular trafficking was microtubule-mediated [10]. Additionally, others have demonstrated that myosin Va, a motor protein usually involved in secretory granule trafficking to the PM, was activated in samples from human simplex virus (HSV)-1 infections and was involved in cortical actin-mediate transport of the virions from the TGN to the PM [30]. Although previous electron microscopy studies focused on mature EBV [22,23,24,25,26] and KSHV [28], the mechanisms of their release from host cells remain unclear.

Here, I adapted two independent induction-systems of the EBV lytic cycle in vitro [29,31,32,33,34,35,36,37,38] to the context of Rab8a, Rab10, or Rab11a silencing, to characterize the EBV release pathway. The obtained data suggest that vesicles containing mature EBV are trafficked to the PM taking advantage of the cellular secretory pathway and provide clarity to the release mechanisms of EBV virions.

## 2. Materials and Methods

### 2.1. Cell Culture and EBV Lytic Cycle Induction

EBV-positive Akata (Akata^+^) cells were derived from a Japanese BL patient [31,32,33,34,35,36]. EBV-negative cells (Daudi^−^) were isolated from African BL-derived EBV-positive Daudi cells, by limiting dilution [39]. Akata^+^ and Daudi^−^ cells were maintained in RPMI-1640 medium containing 10% FBS and antibiotics at 37 °C in a 5% CO_2_ atmosphere. Human embryonic kidney 293 cells, which were modified to be an inducible system of EBV lytic infection (i293/2089), were maintained in Dulbecco’s modified eagle medium (DMEM) containing 10% FBS, antibiotics, 1 µg/mL puromycin, and 100 µg/mL hygromycin at 37 °C with a 5% CO_2_ atmosphere. This cell line carries a BACmid coding for the complete EBV genome (B95.8 strain), green fluorescent protein (GFP), and hygromycin B resistance gene [37], and it stably expresses a tamoxifen-inducible EBV-encoded BZLF1 derivative [38].

The EBV lytic cycle was induced in Akata^+^ or i293/2089 cells as previously reported. Akata^+^ cells, plated in 6-well plates (1 × 10^6^ cells/well), were cultured for 48 h in the presence of 1% goat anti-human IgG F(ab’)_2_ polyclonal antibody (αhIgG, Agilent, Santa Clara, CA, USA) [40,41]. i293/2089 cells, plated in 24-well plates (5 × 10^5^ cells/well), were cultured for 48 h with 100 nM 4-hydroxytamoxifen (Sigma-Aldrich, St. Louis, MO, USA) [37,38]. Lytic cycle induction was always confirmed by Western blotting.

### 2.2. siRNA Treatments

To knockdown Rab8a, Rab10, or Rab11a gene expression, Akata^+^ or i293/2089 cells were transfected with siRNAs encoding the corresponding target sequences: 5′-GAGUCAAAAUCACACCGGA-3′ (siRab8a), 5′-GGACGACAAAAGAGUUGUA-3′ (siRab10), or 5′-GAGAUUUACCGCAUUGUUU-3′ (siRab11a) (Thermo Fisher Scientific, Waltham, MA, USA). As a control, a siRNA encoding a mock-sequence (Thermo Fisher Scientific) was used. Transfection was performed using the Neon Transfection System (Thermo Fisher Scientific) for Akata^+^ cells and the TransIT-TKO reagent (Mirus Bio, Madison, WI, USA) for i293/2089 cells, according to the manufacturers’ instructions. Rab GTPases downregulation was confirmed by Western blotting. Transfected Akata^+^ cells were used to evaluate distribution of gp350/220 by immunofluorescence staining. Transfected i293/2089 cells were used for analysis of the EBV virion secretion.

### 2.3. Western Blotting

Western blotting was performed as previously [29]. Induction of the EBV lytic cycle was confirmed by incubation with goat anti-EBV p18 polyclonal antibody (Thermo Fisher Scientific) diluted at 1:1000. Downregulation of Rab proteins was evaluated by incubation with rabbit anti-Rab8a (clone D22D8, Cell Signaling Technology, Danvers, MA, USA), rabbit anti-Rab10 (clone MJF-R23; Abcam, Cambridge, UK), or mouse anti-Rab11 (clone 47/Rab11; BD, Franklin Lakes, NJ, USA) monoclonal antibodies diluted at 1:1000. Signals were acquired using a LuminoGraph II imaging system (Atto corporation, Tokyo, Japan). Band intensity was quantified using the ImageSaver6 software (Atto corporation).

### 2.4. Immunofluorescence Staining

Akata^+^ cells were fixed with 4% paraformaldehyde in phosphate-buffered saline (PBS) for 10 min, permeabilized with PBS containing 0.05% Triton X-100 for 10 min and blocked in PBS containing 1% bovine serum albumin (BSA) for 20 min, all at room temperature (r.t.). Cells were then incubated with mouse anti-EBV gp350/220 monoclonal antibody (clone C-1) [42], rabbit anti-Rab8a monoclonal antibody (clone D22D8), rabbit anti-Rab10 monoclonal antibody (clone MJF-R23), or rabbit anti-Rab11a polyclonal antibody (Abcam) diluted at 1:200 in PBS for 1 h at r.t. After washing in PBS, cells were incubated with Alexa Fluor 594-labeled anti-mouse IgG F(ab’)_2_ and/or Alexa Fluor 488-labeled anti-rabbit IgG F(ab’)_2_ (all from Thermo Fisher Scientific) diluted at 1:1000 in PBS for 1 h at r.t. After washing in PBS, the nuclei were counterstained with Hoechst 33342 (Cell Signaling Technology). Preparations were observed with a 60 × oil-immersion objective (NA = 1.3) using a confocal laser scanning microscope (Fluoview FV10i; Olympus, Tokyo, Japan) and acquired using the FV10-ASW software (Olympus). Images of four randomly selected independent fields containing between 5 and 10 gp350/220-positive cells (per condition) were captured. Line scan imaging and measurements of the integrated fluorescent intensities in defined regions of interest were performed using the FV10-ASW software (Olympus).

### 2.5. Evaluation of EBV Virion Secretion by Flow Cytometry

i293/2089 cell supernatants were harvested after EBV lytic cycle induction and incubated with Daudi^−^ cells (2 × 10^5^ cells/condition) for 1 h at 37 °C. Cells were then washed in culture media and cultured for 48 h. The percentage of GFP-positive cells was then measured by flow cytometry using a FACSCalibur (BD).

### 2.6. Statistical Analysis

The results are representative of three independent experiments. Statistical differences were assessed by the Student’s t-test and determined significant when *p* < 0.05.

## 3. Results

### 3.1. EBV Structural Proteins Partially Co-Localize with Cellular Secretory Vesicles

In living cells, synthesized proteins are properly distributed to the distinct organelles, the PM, or even to the extracellular milieu via secretion. The TGN is a key station of the constitutive secretory pathway, which is responsible for these sorting processes in all cell types [43]. To understand the potential involvement of the secretory pathway in the release of EBV into the extracellular milieu, the viral lytic cycle was induced in Akata^+^ cells by cross-linking their cell-surface IgG molecules [40,41] and the viral progeny intracellular distribution, together with the distribution of three Rab GTPases, were analyzed by immunofluorescence staining. Forty-eight hours post-induction, the observation of one of the major EBV envelope glycoproteins, gp350/220 [44,45], known to be expressed during the late phase of its lytic cycle and exhibiting similar cytoplasmic distribution to the EBV viral capsid antigen-p18 [29], suggested the newly formed virions were mainly localized in the cytoplasm, with a speckled pattern, and also in the PM (magenta; Figure 1).

Looking at Rab8a, a small GTPase that regulates secretory vesicle transport from the TGN to the basolateral PM of epithelial cells, neuronal dendrites, and cilia [46,47], was, as expected in basal conditions (non-induced cells), distributed in the perinuclear region and in the cytoplasm, forming vesicle-like structures (white; Figure 1B). The localization of Rab8a was maintained in cells induced for the EBV lytic cycle (green; Figure 1A) and partially co-localized with gp350/220 staining (magenta; Figure 1A). These observations were overall similar for Rab10, a transport mediator of glucose transporters as well as toll-like receptor 4, to the PM [46,47]. Rab10 also showed perinuclear localization in a more intense fashion, together with a cytosolic distribution associated with vesicle-like structures, both in basal (white; Figure 1D) and induced states (green; Figure 1C). Additionally, in line with the result obtained for Rab8a, gp350/220 partially co-localized with Rab10, both in the perinuclear region and in the cytosol: some gp350/220-positive speckles were frequently seen adjacent to Rab10 signals (Figure 1C; white arrows). On the other hand, Rab11a, a GTPase indispensable for both regulated secretory pathways and constitutive recycling processes [48,49,50], was detected in the perinuclear regions of untreated cells (white; Figure 1F), however, its localization pattern changed with EBV lytic cycle induction. Rab11 signals decreased and scattered in gp350/220-positive cells (green; Figure 1E), in line with previous observations [29], and Rab11-gp350/220 co-localization was less frequently detected. Overall, these results suggest that mature EBV virions are trafficked to the host cells PM within vesicles containing Rab8a, Rab10, and even Rab11a, after acquiring their secondary envelope from Golgi-derived intracellular compartments.

### 3.2. Downregulation of Rab Proteins Promotes the Intracellular Accumulation of EBV Structural Proteins

To further understand the involvement of the host secretory pathway in the release of EBV progeny, the three Rab GTPases were knocked down in Akata^+^ cells and the distribution of EBV glycoprotein was evaluated by immunofluorescence. The knockdown of the target proteins was confirmed by Western blot analysis (Figure 2C). Remarkably, while control siRNA-treated cells (Figure 2A, top middle) showed EBV distribution patterns similar to the ones in untreated cells (Figure 1 and Figure 2A, top left), those in which each of the Rab proteins was downregulated showed accumulation of viral gp350/220 in the cytoplasmic regions and the PM (Figure 2A,B). In line with this result, expression of EBV-encoded capsid antigen (VCA)-p18 (or BFRF3) [51,52] was increased when the EBV lytic cycle was induced in siRNA-treated cells (Figure 3), suggesting the intracellular accumulation of nascent EBV virions. Overall, these results indicate that the downregulation of Rab GTPase proteins impairs the transport of mature EBV from the compartments in the cytoplasm to the extracellular space.

### 3.3. Downregulation of Rab Proteins Inhibits the Release of Infectious EBV Virions

To confirm the role of the secretory pathway in the release of infectious EBV virions, the EBV lytic cycle was induced in i293/2089 transfected with siRNAs targeting each individual Rab protein (Figure 4C). Two days post-induction, the culture supernatants containing recombinant EBV were harvested and incubated with EBV-negative Daudi^−^ cells, and their infection frequencies given by GFP expression were evaluated by flow cytometry. Remarkably, the knockdown of Rab8a, Rab10, and Rab11a significantly inhibited the release of mature virions (Figure 4A,B), as evidenced by the significantly lower number of infected Daudi^−^ cells compared with the counts of infected cells among those treated with the control siRNA (Figure 4B, *p* value). These results indicate that EBV infectious virions are released from the cell compartments where they acquire their final envelopment via the secretory pathway.

## 4. Discussion

Accumulating evidence indicates that herpesvirus subfamilies share a mechanism for the maturation and egress of their progeny virions [2,3,4]. However, the detailed mechanism underlying the release of mature herpesviruses, particularly, the gammaherpesviruses, is not well understood.

This work characterized the mechanism of EBV virion release into the extracellular milieu. Immunofluorescence staining revealed that small GTPases involved in the cellular secretory pathway, such as Rab8a, Rab10, and Rab11a, partially co-localized with a viral glycoprotein (Figure 1). Furthermore, the knockdown of each of these three Rab proteins promoted the intracellular accumulation of EBV structural proteins (Figure 2 and Figure 3) and suppressed the release of infectious EBV virions (Figure 4). These results support a model in which EBV exploits the host secretory pathway to ensure the release of progeny virions into the extracellular space (Figure 5).

Because Rab GTPases control vesicle budding and fusion events, and because of their motility via the recruitment of effectors, such as sorting adaptors, tethering factors, enzymes, and motor proteins, they have been used to understand the dynamics of intracellular membrane trafficking [53]. However, the associations of the Rab-effector proteins are transient and dynamic. In fact, Rab protein conversion appears to mediate endosomal maturation [54]. Therefore, the observed partial co-localizations of EBV structural proteins and Rab GTPases are probably a consequence of this dynamicity. Finally, to understand the detailed molecular mechanisms of EBV maturation and egress processes, further investigations and the use of alternative methods, such as live-cell imaging and super resolution microscopy, are required.

Trafficking of intracellular organelles is tightly regulated by the microtubule cytoskeleton. Various microtubule-binding proteins, responsible for microtubule-dependent intracellular membrane dynamics, were identified. Interestingly, microtubules were reported to be responsible for the intracellular trafficking of nascent alphaherpesviruses-containing vesicles [10,30], suggesting that the egress processes of other herpesvirus subfamilies may also be regulated by the same machinery.

The efficiency of EBV infection in epithelial cells is significantly enhanced in the context of co-cultures [33,35,55], suggesting that cell-to-cell contact-mediated viral transmission is a dominant EBV infection mechanism. Cell-to-cell contact-mediated viral transmission mechanisms are well-characterized in the human immunodeficiency virus (HIV) that establishes a virological synapse (VS). The VS is an actin-dependent and microtubule-dependent stable adhesive junction that allows the effective transfer of retroviruses from infected dendritic or T cells to non-infected target-cells in the absence of cell-to-cell fusion events [56,57]. Interestingly, HIV is also known to exploit the secretory pathway of CD4^+^ T cells to polarize the distribution of viral antigens towards the VS [58]. However, contrarily to retroviruses, in EBV this viral transmission mechanism is VS-independent [35]. We have shown that cell contact induces multiple signal-transduction pathways, resulting in the initiation of EBV replication [33]. We also found that EBV exploits the pre-existing host endocytic machinery for the establishment of efficient viral transmission. In particular, we observed that cell contact facilitated the recycling of adhesion molecules to the cell surface of EBV-positive B cells, leading to the stabilization of cell contact in a Rab11-dependent manner. However, because the mechanisms that direct newly formed EBV virions towards the sites of contact between infected and adjacent cells are poorly understood, further investigation of the secretory pathway role in this process is essential.

## 5. Conclusions

Taken together, the findings suggest that mature EBV virions, with secondary envelopes derived from the Golgi apparatus, are released into the extracellular milieu via the secretory pathway, as well as providing new insights into the EBV life cycle in general.

## Figures and Tables

**Figure 1 microorganisms-08-00729-f001:**
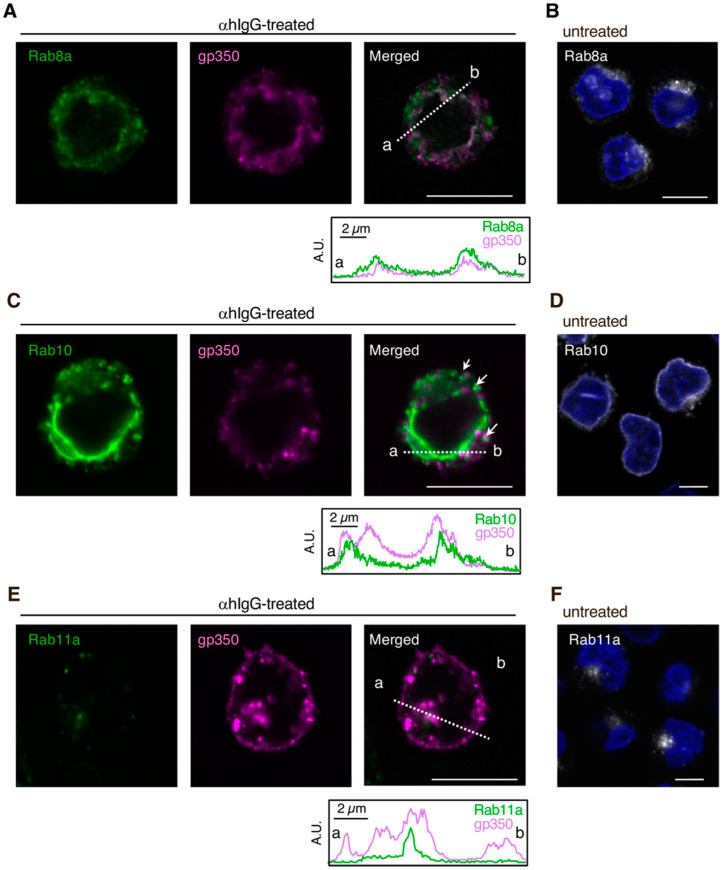
Epstein-Barr virus (EBV) glycoprotein localized in the compartments containing the markers of the secretory pathway. The distribution of EBV glycoproteins and Rab8a (**A**), Rab10 (**C**) or Rab11a (**E**) in Akata^+^ cells undergoing the lytic cycle. Akata^+^ cells were treated with or without αhIgG for 48 h. The distribution of Rab proteins (green), gp350/220 (magenta), and merged images are shown. As a control, the distribution of Rabs (white) in the untreated cells is shown (**B**,**D**,**F**). The nuclei (blue) were counterstained with Hoechst 33342. The plots indicate the individual fluorescence intensity along each of the corresponding lines. A.U., arbitrary unit. Scale bars: 10 µm.

**Figure 2 microorganisms-08-00729-f002:**
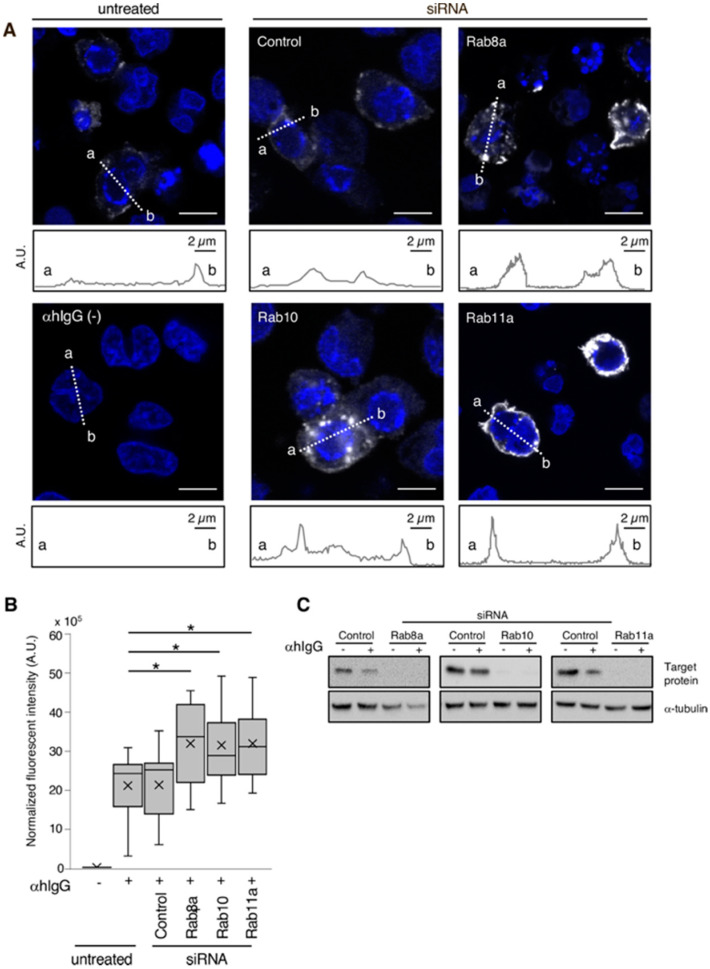
The effect of Rab GTPase downregulation in EBV glycoprotein distribution. Rab8a, Rab10 and Rab11a were downregulated in Akata^+^ cells using siRNAs. After 48 h of treatment with αhIgG for the EBV lytic cycle induction, the distribution of EBV glycoprotein gp350/220 was analyzed by immunofluorescence staining. (**A**) Images of gp350/220-specific signals (white) are shown. As a control, the images of the siRNA-untreated cells with (top left) or without (bottom left) the lytic cycle induction are shown. Cell nuclei were counterstained with Hoechst 33342 and appear in blue. Plots indicate the fluorescence intensity along each of the corresponding lines. A.U., arbitrary unit. Scale bars: 10 µm. (**B**) Summary of gp350/220 expression. Boxplots represent the fluorescent intensities from gp350/220-positive cells in four independent randomly selected fields. Results are representative of three times independent experiments. Average (a cross), median (a line), and standard deviations values are shown. Statistical differences were assessed using Student’s t test and are represented as * *p* < 0.05 versus respective control. (**C**) Downregulation of Rab proteins was confirmed by Western blot.

**Figure 3 microorganisms-08-00729-f003:**
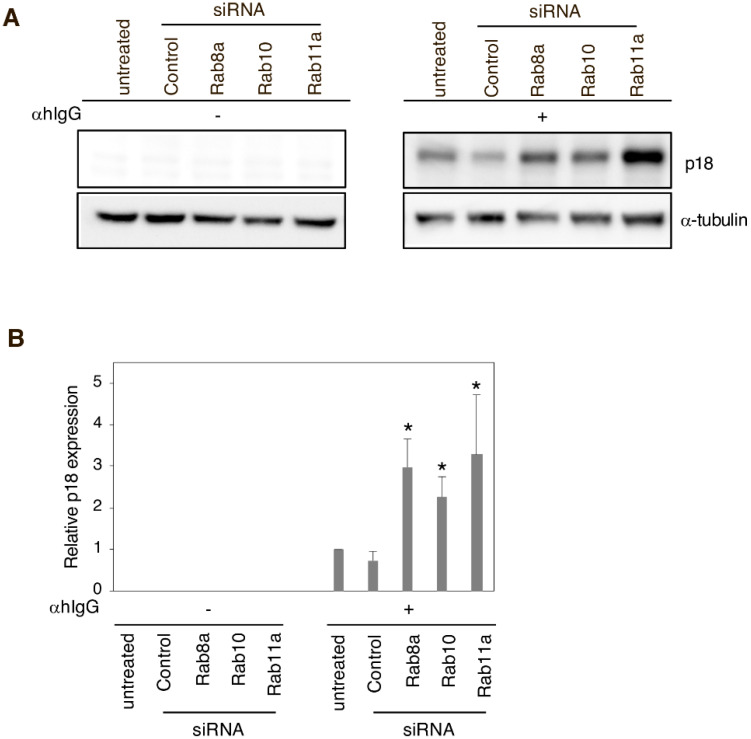
The effect of Rab GTPase downregulation in EBV-encoded capsid antigen p18 expression. Rab8a, Rab10, and Rab11a were downregulated in Akata^+^ cells using siRNAs. After 48 h of treatment with αhIgG for the EBV lytic cycle induction, and the semi-quantification of the EBV-encoded capsid, antigen p18 was performed by Western blot. (**A**) The representative data are shown. (**B**) Normalized p18 expression values to the reference α tubulin are shown. Results are representative of independent experiments conducted three times and the average and standard deviations values are shown. Statistical differences were assessed using Student’s t test and are represented as * *p* < 0.05 versus respective control.

**Figure 4 microorganisms-08-00729-f004:**
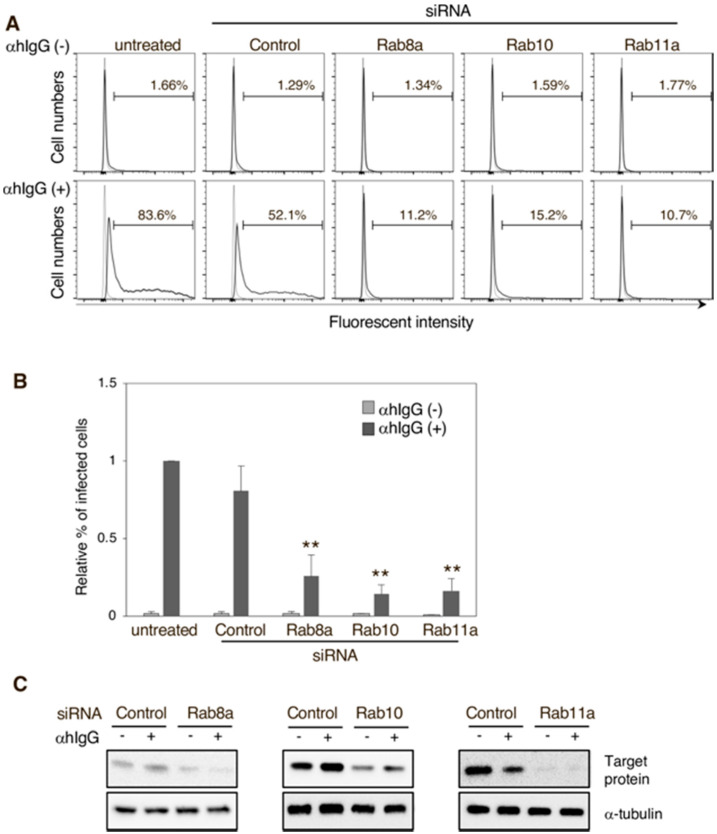
The effect of Rab GTPase downregulation on the release of infectious EBV virions. i293/2089 cells transfected with or without Rab8a, Rab10, or Rab11a siRNAs were treated with tamoxifen for 48 h to induce the EBV lytic cycle. Their supernatants were then harvested and added to Daudi^−^ cells. Two days later, the percentage of GFP-positive Daudi^−^ cells was analyzed by flow cytometry. (**A**) Representative histograms are shown. As a control, Daudi^−^ cells were incubated without the supernatants (thin lines). (**B**) The average infection frequencies and standard deviations are also represented. Results reflect three independent experiments. Statistical differences were assessed using Student’s t test and are represented as ** *p* < 0.01. (**C**) Downregulation of Rab proteins was confirmed by Western blot.

**Figure 5 microorganisms-08-00729-f005:**
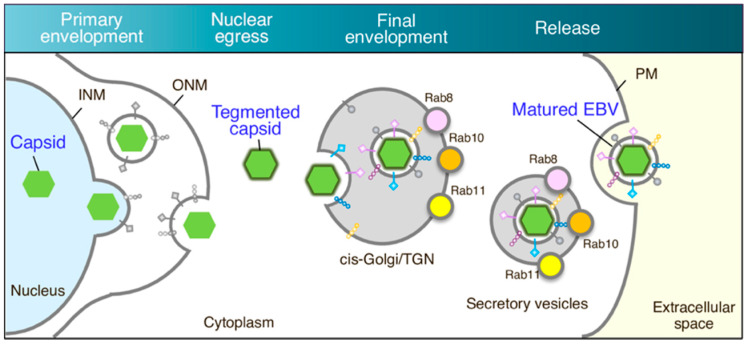
EBV virion maturation model. Replicated viral DNAs are packaged into capsids in the nucleoplasm. Nucleocapsids acquire primary envelopes by budding through the inner nuclear membrane (INM) into the perinuclear space. Perinuclear enveloped virus particles undergo de-envelopment, mediated by the fusion of the primary envelope with the outer nuclear membrane (ONM) (nuclear egress). Tegument-coated nucleocapsids then undergo a final envelopment step by budding into intracellular compartments derived from the cis-Golgi and trans-Golgi networks, producing mature virions. Rab8a, Rab10, and Rab11a-positive vesicles containing mature virions are then transported to the cell surface and fuse with the PM to release EBV virions into the extracellular milieu.

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
