# Peer review of "Epstein-Barr Virus Exploits the Secretory Pathway to Release Virions"

_microorganisms, 2020, doi:10.3390/microorganisms8050729_

Round 1

Reviewer 1 Report

This study examines the role of three Rab GTPases in the release of EBV virions from infected cells.

The author shows that expression of the EBV enevlope glycoprotein, gp350/220, shows partial localisation with these Rab GTPases, less convincingly so for Rab11a. These results suggest that mature EBV virions are trafficked to the plasma membrane in vesicles containing these Rab GTPAses. However, additional clarification is required here- what is the evidence that gp350/220 is a good marker of virions in cells? is it not possible that gp350/220, perhaps when newly formed, is not associated with virions? Is it not possible to visualise virion DNA directly? This should at least be commented on.

The knockdown of each of the Rab GTPases lead to a focal redistribution of EBV virion proteins, suggesting accumulation of virions and a role for the Rab GTPases in the transport of virions. Down-regulation of Rab GTPases also reduced the amount of infectious virus in the supernatant of cells, supporting a role for these celluar proteins.

Minor:

The statement at the beginning of Section 3.2 'To completely understand the involvement of the host cell secretory pathey in the release of EBV progeny' could not be attained in this study and should be revised

Author Response

Dear Reviewer 1,

Thank you for your decision and these thoughtful comments, which I have addressed by modifying the main text. I hope that the manuscript is now acceptable for publication in Microorganisms and thank you for your consideration.

Sincerely yours,

Asuka Nanbo, Ph.D.

A few specific comments for the authors' consideration are listed below.

The author shows that expression of the EBV enevlope glycoprotein, gp350/220, shows partial localisation with these Rab GTPases, less convincingly so for Rab11a. These results suggest that mature EBV virions are trafficked to the plasma membrane in vesicles containing these Rab GTPAses. However, additional clarification is required here- what is the evidence that gp350/220 is a good marker of virions in cells? is it not possible that gp350/220, perhaps when newly formed, is not associated with virions? Is it not possible to visualise virion DNA directly? This should at least be commented on.

Regarding the reviewer 1’s concern, we have previously demonstrated that gp350/220 and the EBV capsid antigen-p18 similarly co-localized with a maker for the Golgi apparatus, suggesting that mature virions bud into the cellular compartments derived from the Golgi body (Nanbo et al, 2018). I have modified the text to be “one of the major EBV envelope glycoproteins, gp350/220, known to be expressed during the late phase of its lytic cycle and exhibited similar cytoplasmic distribution to the EBV viral capsid antigen-p18, suggested the newly formed virions were mainly localized in the cytoplasm, with a speckled pattern, and also in the PM” (page 3, line 144-page 4, line 147).

Minor:

The statement at the beginning of Section 3.2 'To completely understand the involvement of the host cell secretory pathey in the release of EBV progeny' could not be attained in this study and should be revised

Based on the reviewer 1’s comment, I have revised the sentence to be “To further understand the involvement of the host secretory pathway in the release of EBV progeny” (page 5, lines 206-207).

Reviewer 2 Report

Dear Author,

Your work, entitled "Epstein–Barr Virus exploits the secretory pathway to release virions" presents new interenting findings and suggests that mature EBV virions, with secondary envelopes derived from the Golgi apparatus, are released into the extracellular milieu via the secretory pathway. Therefore you provide new insights into the EBV life cycle in general.

I am really impressed that you were able to prepare your article alone as you were solely responsible for: conceptualization, methodology, validation, formal analysis, visualization, investigation, resources, data curation, project administration, funding acquisition, writing—original draft preparation and writing—review and editing.Congratulations!

I have only one suggestion: please add some new references, published during last two years. For example:

-Criscitiello, M.F.; Kraev, I.; Lange, S. Post-Translational Protein Deimination Signatures in Serum and Serum-Extracellular Vesicles of Bos taurus Reveal Immune, Anti-Pathogenic, Anti-Viral, Metabolic and Cancer-Related Pathways for Deimination. Int. J. Mol. Sci. 2020, 21, 2861.

-Münz, C. The Role of Dendritic Cells in Immune Control and Vaccination against γ-Herpesviruses. Viruses 2019, 11, 1125.

-Byrne, A.; Bush, R.; Johns, F.; Upadhyay, K. Limited Utility of Serology and Heterophile Test in the Early Diagnosis of Epstein–Barr Virus Mononucleosis in a Child after Renal Transplantation. Medicines 2020, 7, 21.

-Imai, K.; Ogata, Y. How Does Epstein–Barr Virus Contribute to Chronic Periodontitis? Int. J. Mol. Sci. 2020, 21, 1940.

Author Response

Dear Reviewer 2,

Thank you for your decision and these thoughtful comments, which I have addressed by modifying the main text. I hope that the manuscript is now acceptable for publication in Microorganisms and thank you for your consideration.

Sincerely yours,

Asuka Nanbo, Ph.D.

Your work, entitled "Epstein–Barr Virus exploits the secretory pathway to release virions" presents new interenting findings and suggests that mature EBV virions, with secondary envelopes derived from the Golgi apparatus, are released into the extracellular milieu via the secretory pathway. Therefore you provide new insights into the EBV life cycle in general.

I am really impressed that you were able to prepare your article alone as you were solely responsible for: conceptualization, methodology, validation, formal analysis, visualization, investigation, resources, data curation, project administration, funding acquisition, writing—original draft preparation and writing—review and editing.Congratulations!

I have only one suggestion: please add some new references, published during last two years. For example:

-Criscitiello, M.F.; Kraev, I.; Lange, S. Post-Translational Protein Deimination Signatures in Serum and Serum-Extracellular Vesicles of Bos taurus Reveal Immune, Anti-Pathogenic, Anti-Viral, Metabolic and Cancer-Related Pathways for Deimination. Int. J. Mol. Sci. 2020, 21, 2861.

-Münz, C. The Role of Dendritic Cells in Immune Control and Vaccination against γ-Herpesviruses. Viruses 2019, 11, 1125.

-Byrne, A.; Bush, R.; Johns, F.; Upadhyay, K. Limited Utility of Serology and Heterophile Test in the Early Diagnosis of Epstein–Barr Virus Mononucleosis in a Child after Renal Transplantation. Medicines 2020, 7, 21.

-Imai, K.; Ogata, Y. How Does Epstein–Barr Virus Contribute to Chronic Periodontitis? Int. J. Mol. Sci. 2020, 21, 1940.

I appreciate the reviewer 2’s suggestion. I have added the following three relevant references published in last two years.

Muller YA et al. High-resolution crystal structures of two prototypical β- and γ-x unravel the determinants of subfamily specificity. J Biol Chem. 2020 Mar 6;295(10):3189-3201.

Yanagi Y  et al. Initial Characterization of the Epstein-Barr Virus BSRF1 Gene Product. Viruses. 2019 Mar 21;11(3).

Dai YC et al. The Novel Nuclear Targeting and BFRF1-Interacting Domains of BFLF2 Are Essential for Efficient Epstein-Barr Virus Virion Release. J Virol. 2020 Jan 17;94(3). pii: e01498-19.